# Surveillance of Antimicrobial Resistance of *Escherichia coli* Isolates from Intestinal Contents of Dairy and Veal Calves in the Veneto Region, Northeaster Italy

**DOI:** 10.3390/ani14101429

**Published:** 2024-05-10

**Authors:** Laura Bortolami, Antonio Barberio, Eliana Schiavon, Federico Martignago, Erica Littamè, Anna Sturaro, Laura Gagliazzo, Alessia De Lucia, Fabio Ostanello

**Affiliations:** 1Istituto Zooprofilattico Sperimentale delle Venezie, Viale dell’Università, 10, Legnaro, 35020 Padova, Italy; lbortolami@izsvenezie.it (L.B.); abarberio@izsvenezie.it (A.B.); eschiavon@izsvenezie.it (E.S.); fmartignago@izsvenezie.it (F.M.); elittame@izsvenezie.it (E.L.); asturaro@izsvenezie.it (A.S.); lgagliazzo@izsvenezie.it (L.G.); 2Azienda ULSS 3, Via 29 Aprile, 2, Dolo, 30031 Venezia, Italy; alessia.delucia@aulss3.veneto.it; 3Department of Veterinary Medical Sciences, University of Bologna, Via Tolara di Sopra, 50, Ozzano dell’Emilia, 40064 Bologna, Italy

**Keywords:** *Escherichia coli*, antimicrobial resistance, AMR, multidrug resistance, MDR, minimum inhibitory concentration, MIC, veal calves, dairy calves, Italy

## Abstract

**Simple Summary:**

One key factor in the emergence of antimicrobial resistance (AMR) in microorganisms is antimicrobial usage in veterinary and human medicine. In dairy cattle, the prevalence of AMR *E. coli* is age-dependent, with a higher prevalence detected during the early stages of life. In veal calf fattening herds, the high degree of commingling calves from different farms of origin can lead to the rapid spread of infections within the farm. This can cause severe health issues, necessitating greater use of antimicrobial agents. Antimicrobial use has been reported to be higher in the veal calf fattening sector compared to the other production branches of livestock production. This represents an unquantified risk to human health, as antimicrobial-resistant bacteria and the genetic elements mediating resistance can potentially spread through foodborne transmission or environmental dissemination. This surveillance study was conducted, using data available from routine diagnostic activity, to: (i) determine the proportion of antimicrobial-resistant strains and AMR profiles of *E. coli* isolated from the intestinal contents of veal and dairy calves in the Veneto Region (Northeaster Italy); (ii) investigate potential variation in the AMR profiles of isolates between the two categories of calves and, (iii) identify potential AMR variations over the period considered (2017–2022).

**Abstract:**

This surveillance study aimed to estimate the proportion of antimicrobial resistant strains and antimicrobial resistance (AMR) profiles of *E. coli* isolates detected from the intestinal contents of veal and dairy calves in the Veneto Region, Northeaster Italy. Additionally, we investigated the differences in AMR profiles between dairy and veal calves over the period 2017–2022. Overall 1150 *E. coli* isolates were tested from calves exhibiting enteric disease, with 868 from dairy and 282 from veal calves. The percentage of resistant isolates to nine antimicrobials was notably higher in veal calves compared to dairy calves, except for ampicillin. Throughout the study period, we observed a significant increase in the proportion of resistant isolates to florfenicol, gentamycin, paromomycin, tetracycline and trimethoprim/sulfamethoxazole in dairy calves, while we did not detect any significant increase in the proportion of resistant isolates among veal calves. A substantial proportion (75.9%) of the isolated *E. coli* exhibited multi-drug resistance (MDR). The proportion of multi-drug resistant isolates was significantly higher in veal calves (91.7%) compared to dairy calves (74.3%) all through the surveillance period (2017–2022), with no significant variation in MDR proportion among veal calves between 2017 and 2022 but a significant increase among dairy calves.

## 1. Introduction

Antimicrobial resistance (AMR) is a naturally occurring phenomenon resulting from the evolutionary adaptation of bacteria [1,2]. Nowadays, it is recognized as a global threat to public health, necessitating a One Health approach to tackle its spread [3]. The proportion of AMR in bacteria has increased dramatically over the past few decades, posing a critical threat to both the global public and animal health [2]. Antimicrobial use (AMU) in veterinary and human medicine is a key factor in the emergence, selection, and dissemination of antimicrobial-resistant microorganisms [1]. The use of antimicrobials in livestock is recognized as a key driver behind the escalation of AMR, affecting not only pathogens but also indicator bacteria [4], such as *Escherichia coli*, a ubiquitous microorganism that can behave both commensally and as a pathogen [5].

In food-producing animals like veal calves, resistant microorganisms can be introduced onto farms from outside sources, such as new livestock, vectors including rodents, birds, and insects, or through contaminated feed and water [6]. Furthermore, antimicrobial-resistant bacteria can disseminate resistance genes across diverse microbial communities via mobile genetic elements, such as plasmids and transposons [6]. 

According to official data, antibiotic resistance in cattle is a serious issue in Italy. This country has the lowest proportion of fully susceptible *E. coli* in bovines under one year of age [7]. Moreover, the combined resistance to 3rd generation cephalosporins and fluoroquinolones is also very frequent as Italy ranks second after North Macedonia among European countries. No decreasing trend was observed for these resistance indicators in cattle in Italy in the last years [7].

Despite a remarkable decrease in AMU in food animal production in Italy in recent years [8], it remains uncertain whether this decline has affected the proportion of AMR on farms. This research aims to fill this gap, providing crucial data for responsible AMU practices and detecting resistance emergence [9], while supplying veterinarians with data to optimize therapy. 

*Escherichia coli* isolates are chosen for their suitability in measuring antimicrobial resistance across various ecological niches, including both human and animal settings [9]. Moreover, *E. coli* serves as an excellent sentinel for AMR across a wide range of species and is considered a potential reservoir of resistance genes that could transfer resistance to other zoonotic or commensal organisms, posing risks to both animals and human health [10].

In dairy cattle, the prevalence of antimicrobial-resistant *E. coli* and antimicrobial resistance genes (ARGs) is influenced by age, with a higher prevalence detected in early life stages, especially among calves of 2–4 weeks old [11,12,13], compared to other categories in cattle breeding [14]. Several studies examining the high prevalence of antimicrobial-resistant *E. coli* in young dairy calves have yielded evidence of milk diet as a significant risk factor [12,13,15,16]. Moreover, the supplementation of calf milk with antimicrobials, the incidence of diarrhoea, the use of feed additives with biocides and heavy metals, as well as vitamin supplements have also been suggested to contribute to the high AMR prevalence in young calves [12,15,17]. Previous studies have suggested that maternal colostrum, as the initial feed, might be a significant vehicle for antimicrobial-resistant *E. coli* transmission in neonatal calves [15,18]. 

Given the potential negative impact on meat production and public health, investigating AMR trends in bovine production in Europe, particularly in countries like Italy with a long and rich tradition of animal husbandry, is essential [19]. In 2019, the EU-27 produced approximately 645,000 tons of veal meat from 4.4 million calves, with major production countries including The Netherlands (36%), France (28%), Italy (13%), Belgium (9%), and Germany (7%) [20].

Surplus dairy calves refer to those born on dairy farms that are either unsuitable or unnecessary for replacing the milking herd. These calves are mostly male and are typically sold for grain-fed veal production, during which they are raised primarily on a milk-based diet until 6–8 weeks of age before transitioning to a grain-based diet and being marketed at 8 months of age [21].

In Italy, veal calves usually enter the fattening unit around three weeks of age and are fattened until they reach an average live body weight of kg 270 (around 7–8 months of age), adhering to European Union welfare regulations [22]. 

The production stages in surplus calves involve frequently long-distance transportation to livestock markets, where they are purchased by calf raisers for rearing. Throughout these stages, surplus calves experience many health challenges [23,24,25], increasing the risk of infectious disease and subsequent AMU [26], especially in the first weeks after housing. Due to the high degree of commingling of calves from different farms’ origin at these markets, infected calves can lead to a rapid spread of disease on the veal calf farm, causing severe health issues resulting in increased AMU, including preventive use of antimicrobials [27], and economic losses [28]. Preventive antimicrobial treatment has been restricted in the European Union by guidelines and regulations [29,30]. However, the arrival of sick animals at fattening units is common, and antimicrobial group treatment of healthy but potentially infected animals alongside sick calves is largely practiced [31,32]. AMU has been reported to be higher in the veal calf fattening sector rather than in other production branch [33], posing an unquantified risk to human health due to the potential transmission of antimicrobial-resistant bacteria and ARGs through foodborne pathways or environmental dissemination [21]. Previous Italian studies investigating *E. coli* resistance expression in calves have revealed a significant proportion of AMR strains, particularly resistant to ampicillin, tetracycline, and sulfamethoxazole/trimethoprim [5,34]. This trend seems to persist despite a substantial reduction in the usage of diverse antibiotic agents in recent years [8].

Due to the great variability of production systems and producers among countries, standardized AMR data collection for timely and regional comparisons is recommended [35,36]. 

The rearing of beef and dairy calves has fundamental differences due to animal origin, management, type of disease, feeding type, etc. These factors influence the nature and prevalence of infections and, as a consequence, treatment choices. This survey aimed to highlight any differences in the antimicrobial resistance profiles of *E. coli* strains isolated from the intestinal contents of these two categories of calves. To the best of our knowledge, there are no scientific papers available that use standardized methods for surveying AMR data between calves belonging to the two different herd categories (dairy and veal calves) within the same geographical area.

For these reasons, this surveillance study was conducted, using routinely available data, with the following aims: (i) estimating the proportion of antimicrobial resistant strains and AMR profiles of *E. coli* isolated from the intestinal contents of veal and dairy calves in the Veneto Region (Northeaster Italy); (ii) investigating potential differences in the AMR profiles of *E. coli* isolated from the two calf categories; (iii) identifying potential AMR variations over the considered period (2017–2022).

## 2. Materials and Methods

### 2.1. Study Area and Samples Tested

The study was conducted using the data of the antimicrobial susceptibility test (AST) performed between 2017 and 2022 by the Istituto Zooprofilattico Sperimentale delle Venezie (IZSVe), an institution belonging to the Italian network of state veterinary laboratories. Among the ASTs performed for the routine diagnostic, all those were selected executed on *E. coli* isolates collected from dead calves (age 1–180 days) belonging to dairy or veal beef farms, delivered to the laboratory for necroscopy, with a pathological diagnosis of enteritis. The isolates were obtained by bacteriological culture performed on intestinal contents collected during the necroscopy. Only one AST was performed by the *E. coli* isolated from intestinal content for each calf included in the study. The overall amount of *E. coli* isolates ASTs used for this study was 1150. The isolates were collected from 874 dairy calves (76.0%) and 276 veal calves (24.0%), belonging to 429 herds, 333 dairy (77.6%), and 96 veal farms (22.4%) situated in the Veneto Region (Northeaster Italy). The breed of the calves included in the study was pure Italian Holstein or crossbreed (Belgian Blue cross on Italian Holstein). The Veneto Region, spanning an area of 18,345 km^2^, is the third Italian region in terms of cattle population, with approximately 750,000 cattle, of which 64% are dairy cows. This region is home to 2658 dairy herds and holds the second-highest number of Italian farms (over 1000) engaged in rearing veal beef calves (approximately 125,000 calves) [37].

### 2.2. Isolation of E. coli

The isolation procedure remained consistent during the whole study period (2017–2022). The samples were processed within 24 h after collection, cultured on both MacConkey agar plates and blood agar plates, and incubated aerobically for 16 ± 2 h at 37 ± 2 °C. After overnight incubation, suspicious *E. coli* colonies were identified by morphology and Gram staining. For each sample, a single suspected colony with specific biochemical properties (lactose and indole positivity; negativity for H_2_S, oxidase, and urease) were sub-cultured on brain heart infusion (BHI) agar, while identities were confirmed using the API 20E biochemical method (bioMérieux, Marcy l’Etoile, France) according to producer’s instructions [5]. All the media, except API 20E, were produced by the IZSVe internal service for reagents production (IZSVE CSP) starting from lyophilized product of MacConkey (Biomedical service, Scorzè, Italy) and Blood agar base 2 (Biolife Italiana, Milano, Italy) supplemented with sheep defibrinated blood. Production and quality control were performed according to ISO 9001:2015 [38].

### 2.3. Antimicrobial Susceptibility Testing

The Minimum Inhibitory Concentration (MIC) of the nine selected antimicrobials (Table 1) was determined using the broth dilution test according to the procedure described in the Clinical and Laboratory Standards Institute (CLSI) guidelines VET01-5th edition [39].

MIC was evaluated with a customized commercial microdilution MIC system. From 2017 until 2020, MIC plates and Cation Adjusted Mueller-Hinton Broth (CAM-HB) were supplied by Merlin Diagnostika, GmbH (Bornheim, Germany), and from 2021 to 2022 by TREK Diagnostic Systems (East Grinstead, UK).

Both the microdilution MIC systems were produced according to ISO 13485:2016 [45] and each MIC plate batch underwent to quality control with *E. coli* ATCC 25922 and *Pseudomonas aeruginosa* ATCC 27853 strains before releasing. According to CLSI guidelines [39], a pure colony from overnight growths of *E. coli* isolates was picked up with a sterile loop and suspended in sterile saline (IZSVe CSP). The bacterial suspension was adjusted using a nephelometer (Biosan, Riga, Latvia) until an optical density (OD) of 0.5 nephelometric turbidity units (NTU) on the McFarland scale was achieved. Then 50 µL of the suspension were transferred in 11 mL of CAM-HB to provide an inoculum concentration of approximately 10^5^ CFU/mL, and 100 µL of the CAM-HB were transferred to each well of the plate. After incubation at 34 ± 1 °C for 18–24 h, bacterial growth was assessed visually or using a Sensititre Vizion instrument (Thermo Scientific, Loughborough, UK), and the last concentration of antimicrobial that did not show turbidity or a deposit of cells at the bottom of the well was recorded. The MIC value of each isolate, expressed as µg/mL, was defined as the lowest concentration of the antimicrobial agent that completely inhibited the growth after the incubation period and interpreted using clinical breakpoints (CBPs) reported in Table 1. Isolates were differentiated between resistant (R), intermediate (I) and susceptible (S) based on the CBPs. The laboratory selected the antimicrobials based on their activity against cattle enteric pathogens and on their registration by the Italian Ministry of Health (IMH). Due to the change of MIC plates that occurred during the time of observation, only the antimicrobials present in both the MIC plates were included. The antimicrobials included in the study were: ampicillin (AMP), colistin (COL), enrofloxacin (ENR), florfenicol (FLO), flumequine (FLQ), gentamicin (GEN), paramomycin (PAR), tetracycline (TET), trimethoprim/sulfamethoxazole (SX-T). AMP, FLO, GEN, PAR, TET, and SX-T are widely used for therapy in calves. The use of COL and ENR has been restricted to specific cases by the IMH since 2017, so they were included to assess the situation of AMR toward these specific antimicrobials during the time of the study. Eventually, FLQ was included, despite the fact that this antimicrobial is not registered for cattle, because this antimicrobial is a 1st generation quinolone and the results of AMTs can be compared with those obtained by ENR, a 2nd generation quinolone.

Results were interpreted using available CLSI resistance CBPs according to VET08 4th edition guidelines [40], the European Committee on Antimicrobial Susceptibility Testing (EUCAST) guidelines [41], the Comitè de l’Antibiogramme de la Sociètè Française de Microbiologie (CASFM) guidelines [42] and the breakpoints reported in the literature [43] when specific standards were not established by any international recognized guidelines. The criteria used for the selections of the CBPs were cattle, when available, human, and other animal species BPs. The BPs adopted by the laboratory have not been changed by the new updated version of CLSI, CASFM, and EUCAST guidelines, except for ampicillin, for which the resistance BP was lowered from 1 to 0.25 µg/mL by the 2023 CLSI VET 01S edition [46]. Due to the dilution design of the plate (last dilution 0.25 µg/mL), and the small number of misclassified isolates, it was decided to assess the ampicillin according to the previous CBPs. Results were also assessed using the epidemiological cut-off values (ECOFFs) provided by EUCAST [44], except for paramomycin, because ECOFF was not available for this antimicrobial.

For each antimicrobial the median MIC (MIC_50_) and the 90th percentile (MIC_90_) were calculated.

In subsequent analyses, multi-drug resistance (MDR) was assessed according to the definition of resistance to three or more antimicrobial classes [47,48].

### 2.4. Statistical Analyses

Temporal trends in the percentages of antimicrobial and multidrug-resistant *E. coli* were analysed using Linear-by-Linear Cochran-Armitage test [49,50]. A Chi-square test was also applied to investigate the proportion of antimicrobial and multidrug-resistant *E. coli* isolates between two different animal typologies (veal vs. dairy calves). Confidence intervals (95%CI) were calculated using Wall’s method. 

Correlation analysis between year of isolation and the mean MICs value for each antimicrobial tested were performed using Spearman’s *rho* ranked coefficient test.

Statistical analyses were performed using the software SPSS 28.0 (IBM SPSS Statistics, New York, NY, USA), and *p* < 0.05 was set as statistically significant.

## 3. Results

Over the six years surveillance period (2017–2022), the MIC values of 1150 *E. coli* isolates from intestinal contents of dead veal and dairy calves with a pathological diagnosis of enteritis in the Veneto Region (Italy) were investigated. Among these, 874 (76.0%) originated from calves raised in 333 dairy herds and 276 (24.0%) from 96 veal herds. The number of isolates collected from the same herd in the six years ranged from 1 to 25 samples/herd (median = 2) in dairy and from 1 to 33 samples/herd (median = 2) in veal farms.

### 3.1. Proportion of Antibiotic Resistant E. coli

A description of antimicrobials tested with their clinical breakpoints (CBPs) and epidemiological cut-off (ECOFF) is reported in Table 1. Interestingly, the resistance CBP values were greater than the resistance ECOFF values for ENR, FLQ, GEN, SX-T; for AMP and FLO an opposite situation was found with resistance ECOFF values greater than the CBP values. Equal cut-off values were reported for COL and TET. ECOFF value was not available for PRM.

The proportions of *E. coli* isolates resistant to the tested antimicrobial, distinct by the type of breeding, are reported in Table 2. The percentage of isolates resistant to each tested antimicrobial was constantly higher in veal calves than in dairy calves. According to CBPs, those differences were statistically significant (*p* < 0.001) for all the antimicrobials except AMP. Regarding this latter antimicrobial, *E. coli* isolates collected from dairy calves showed the highest resistance rate with only one susceptible and four intermediate isolates. Using ECOFF, a highly significant difference (*p* < 0.001) was also found for ampicillin.

Examining all the results through the entire surveillance period (2017–2022) (Figure 1), there was a significant increase in the proportion of resistance among isolates for FLO (46.9% vs. 57.6%; *p* = 0.002), GEN (25.0% vs. 41.3%; *p* < 0.001), PRM (55.5% vs. 66.3%; *p* = 0.001), TET (77.3% vs. 83.1%; *p* = 0.033) and SX-T (62.5% vs. 75.6%; *p <* 0.001). However, assessing separately the two types of breeding, for veal calves there were no significant (*p* > 0.05) variations in the proportion of resistance isolates along the time to any of the tested antimicrobials. In contrast, a significant change in AMR was observed in dairy calves for GEN (19.8% vs. 37.8%; *p <* 0.001), SX-T (47.7% vs. 72.0%; *p* < 0.001), PRM (45.3% vs. 61.5%; *p <* 0.001), FLO (33.7% vs. 53.1%; *p <* 0.001), and TET (66.3% vs. 81.1%; *p <* 0.001).

### 3.2. Intestinal Carriage of Multidrug-Resistant E. coli in Veal and Dairy Calves

Overall, 902 out of 1150 (78.4%) examined *E. coli* isolates exhibited MDR. Within the MDR category, the median value of resistances was 6 and the maximum value was 8. Only four isolates (0.35%) were sensible to all nine tested antimicrobials. Of the 902 multidrug-resistant isolates, the most prevalent phenotype in both veal and dairy calves was resistance to PRM-AMP-ENR-FLO-FLQ-GEN-PRM-TET-SX-T (66 and 78 isolates respectively), Table 3.

All through the surveillance period (2017–2022), the proportion of MDR isolates in veal calves (91.7%) was significantly higher (*p* < 0.001) than in dairy calves (74.3%). A significant increase in the proportion of all MDR isolates and in proportion of MDR isolates collected from dairy calves was observed (all isolates: from 75.8% in 2017 to 83.7% in 2022, *p* = 0.002; isolates from dairy calves: from 66.3% in 2017 to 81.1% in 2022, *p* < 0.001), Figure 1.

In contrast, no significant variation in the proportion of MDR *E. coli* collected from veal calves was found between the 2017 and 2022 (from 95.2% in 2017 to 96.6% in 2022; *p* = 0.640), Figure 1.

### 3.3. MIC_50_ and MIC_90_ Values of E. coli Isolated Strains Detected in Dairy and Veal Calves

The MIC values of the nine antimicrobial agents obtained from the examinations of *E. coli* isolates are reported in Table 4 and Figure 2. 

A classical unimodal distribution of MIC values was observed for AMP. Specifically, in isolates from dairy calves, 72.5% of the isolates exhibited MIC values surpassing the highest antibiotic concentration on the plate (32 μg/mL), while this percentage soared to 90.6% for isolates from veal calves. AMP had MIC_50_ and MIC_90_ values that exceeded the highest antimicrobial concentration available on the plate (32 μg/mL).

For COL, the majority of isolates (76.4% in veal calves and 86.3% in dairy calves) showed MIC values within the middle range of the MIC values distribution (between 0.25 and 1 μg/mL). In isolates collected from veal calves, MIC_50_ and MIC_90_ values were notably separated (1 and 8 μg/mL, respectively); in contrast, in dairy calves these two values were closely aligned (0.5 and 1 μg/mL) and lower than those observed in veal calves.

ENR MIC values displayed a bimodal distribution, particularly pronounced in isolates from dairy calves, with 35.6% of isolates showing MIC values lower or equal to 0.03 μg/mL, and 39.9% of isolates showing MIC values greater or equal to 8 μg/mL. The MIC_50_ and MIC_90_ values were, respectively, 8 and >32 μg/mL for veal calves and 0.5 and 32 μg/mL for dairy calves.

FLO MIC values tended to concentrate at medium-low concentration levels (ranging from 4 to 16 μg/mL) and showed values exceeding the highest antimicrobial concentration on the plate (64 μg/mL); this distribution was observed in 40.9% and 53.6% on veal calves and 66.8% and 30.3% on dairy calves, respectively.

The MIC values of FLQ were characterized by a bimodal distribution, with 37.2% of isolates from dairy calves showing MIC values ≤ 1 μg/mL and 31.8% with values greater than 32 μg/mL. In veal calves, these percentages were 21.0% and 44.2%, respectively.

Regarding the GEN MIC values, the distribution was uniform for isolates from veal calves, with slightly higher percentages of isolates at the two extremes of antimicrobial concentration. Conversely, in dairy calves, 66.6% of strains had MIC values less than or equal to 1 μg/mL, while 24.1% had values greater than or equal to 16 μg/mL, indicating a bimodal distribution. In strains from both animal categories, MIC_50_ and MIC_90_ values showed considerable separation (veal calves: MIC_50_ = 4 and MIC_90_ > 32 μg/mL; dairy calves MIC_50_ = 0.5 μg/mL and MIC_90_ = 32 μg/mL).

Regarding PRM, about half of the isolates (61.6% in veal calves and 50.2% in dairy calves) showed MIC values exceeding the highest antibiotic concentration on the plate (32 μg/mL). Most of the other values were concentrated at lower dilutions (≤2 μg/mL) (17.3% and 30.5% in veal and dairy calves, respectively).

TET and SX-T MIC values displayed a similar distribution, with the highest percentages of isolates exhibiting MIC values exceeding the highest concentration of antibiotics on the plate (16 μg/mL). Specifically, 93.5% of isolates from veal calves and 75.6% from dairy calves showed TET MIC values exceeding 16 μg/mL. For SX-T, these percentages were 81.5% and 55.8%, respectively. For both these antimicrobials, the MIC_50_ and MIC_90_ values coincided and were greater than 16 μg/mL.

The temporal distributions of mean and median MIC value for each antimicrobial tested are reported in Appendix A. A significant (*p* < 0.05) positive correlation was observed between MIC values and the year of sampling for all antimicrobials examined but COL and FLQ, for which a significant (*p* < 0.05) negative correlation was detected.

## 4. Discussion

This surveillance study aims to estimate the proportion of antimicrobial-resistant strains and AMR profiles of *E. coli* in veal and dairy calves in Northeaster Italy, identify potential differences in AMR profiles between the two categories, and assess AMR variations over time (2017–2022).

To assess AMR, two categorization methods were employed in this research: CBPs and ECOFFs. CBPs classify MIC values into distinct classes of bacterial susceptibility based on clinical outcomes of antimicrobial treatment, while ECOFFs are based on the mathematical analyses of observed MIC distributions to differentiate wild type and non-wild type isolates. It is important to note that classifications based on ECOFFs are not immediately related to classifications based on CBPs, because an isolate identified as non-wild type may still be clinically susceptible [51]. The use of this double assessment could prove highly beneficial in veterinary medicine for cattle due to the lack of CBPs for each combination of specie—microorganism—disease, often necessitating laboratories to adapt CBPs from human or other species. The comparison among CBPs and ECOFFs highlights that for two antimicrobials, ampicillin and florfenicol, the ECOFFs are greater than the resistance CBP. This event could induce the misclassification of several wild type isolates as resistant to these antimicrobials if only the CBP was applied to detect AMR isolates. It should be underlined that, while an isolate identified as a non-wild type may still be clinically susceptible, a wild-type isolate should not be classified as resistant because it lacks phenotypically detectable acquired resistance mechanisms. Interestingly, while the CBP for florfenicol has been adapted from *Salmonella enterica* serotype Cholerasuis in swine, the one for ampicillin is specific for *E. coli* in cattle but referred to metritis. Moreover, only for two antimicrobials, tetracycline and colistin, the resistance BP matched between CBPs and ECOFFs. These findings underscore that, for AMR assessment in food-producing animals, the use of ECOFFs is more reliable and should be preferred over CBPs. Moreover, when assessing the therapeutic outcome of antimicrobial treatment, antimicrobial susceptibility testing (AST) results using CBPs not specifically designed for the animal species—microorganism—disease combination should be carefully evaluated due to possible bias in the interpretation, and compared with the result obtained using ECOFFs, when available.

This surveillance study examines the relationship between the proportion of resistant and multidrug-resistant *E. coli* isolates among the population of calves (dairy or veal) in a region of Italy with a high density of herds. Appling ECOFFs, the lower percentage of antimicrobial-resistant *E. coli* detected was for COL (9.0%), while the three antimicrobials with the highest proportion of antimicrobial-resistant isolates were TET (80.9%), AMP (78.3%), and SX-T (66.2%). The use of the harmonized ECOFFs allow for comparison with data from the European Union (EU) report on AMR in the year 2020–2021 [52]. The average resistance reported by EU survey in calves for Italy is lower than the one observed in this study for ampicillin (60.6 vs. 78.3%), gentamicin (11.2 vs. 29.4%), colistin (0 vs. 9%), while it is nearly comparable for tetracycline (74.7 vs. 80.9%). A possible explanation for this difference could be attributed to the source of samples: diseased animals in this study versus healthy animals and meat products in the EU survey. Calves affected by enteritis show a higher level of resistance compared to healthy ones [53], likely due to the increased AMU in those herds. The resistance levels observed for ampicillin and tetracycline align with the results of other studies conducted on diseased cattle in different European Union countries [53,54]. Furthermore, similar resistance levels were observed in previous surveys conducted in Italy, suggesting that these antimicrobials have the greatest use in the cattle sector [34]. This is also confirmed by the data reported in the Italian national recording system for antimicrobial consumption (ClassyFarm) [55]: in the Veneto region in 2022 aminopenicillins were the second most used antimicrobial in dairy farms, and tetracycline the most used in veal calves. Although the resistance pattern may vary among countries, the high resistance to penicillin, tetracycline and sulphonamides was reported in many different countries both in pathogenic and non-pathogenic isolates of *E. coli* in the EU and all over the world [54,56,57]. Our data, along with earlier studies, support the hypothesis that bacteria of animal origin are commonly resistant to these three antimicrobial classes [50,58]. These results are consistent with the drug consumption reported in the latest ESVAC report by EMA on sales of veterinary antibiotics between 2010 and 2022. Here, it can be observed that penicillin, tetracyclines, and sulfonamides were the most commonly used molecules in Italy in 2022 [8], with a sale for food-producing animals in mg/PCU of 35.6 for tetracyclines, 54.6 for penicillin and 21.8 for sulfonamides.

In a survey on antibiotic usage conducted among farmers and veterinarians for treating neonatal calf diarrhea in Austria, Belgium, Portugal, and Scotland in 2018, the most commonly utilized molecules were quinolones, sulfonamides, and penicillin [59]. An analysis of antibiotic consumption in beef cattle farms in the Veneto region between 2016 and 2019 revealed that the most frequently administered molecules were penicillin (84.4%), followed by quinolones (66.1%), amphenicols (64.0%), macrolides (57.6%), and tetracyclines (40.8%) [60]. Moreover, the ClassyFarm data for the Veneto region in 2022, highlighted that the most used antimicrobials in veal calves were, respectively, tetracyclines, sulfonamides and aminopenicillins. We can, therefore, find some correspondence, at least regarding tetracyclines, penicillin and sulfonamides, between the consumption data reported and the high percentages of resistance expressed by the *E. coli* isolates detected in this study.

The comparison of ECOFFs with the MIC_50_ results allows for assessing the level of *E. coli* resistances detected in the investigated population concerning the average one recorded by the EUCAST ECOFFs. The great differences found between ECOFFs and MIC_50_ for AMP, TET and SXT are consistent with the other outcomes of the study, underling their high usage in calves along the time. In veal calves, the MIC_50_ of ENR was remarkably higher than the ECOFF (8 µg/mL vs. 0.125 µg/mL), while in dairy calves the difference was much lower (0.5 µg/mL vs. 0.125 µg/mL). These outcomes highlight the misuse of this antimicrobial in previous years in veal calves herds and the need to keep strict monitoring on this antimicrobial by the assessment of the consumption and of the dynamic of the AMR. For this purpose, the use of routine AST by MIC method could be a useful tool providing information not only on the percentage of resistant isolates but also on the dynamic of AMR, by monitoring the distribution of MIC values and testing the temporal trends [51]. COL MIC_50_ value showed a completely different scenario compared to the other antimicrobials tested. The overall MIC_50_ and those of veal and dairy calves were lower than the ECOFFs. This outcome is not surprising because acquired resistance by Gram-negative bacteria is rare for COL [61], as confirmed by the low incidence of AMR recorded for this antimicrobial in EU [52]. Moreover, the COL ECOFF is obtained by including human and animal isolates, while ENR ECOFF arises only from animal isolates, because ENR is used only in animals.

In the examination of AMR among various animal types, we observed that veal calves had a higher proportion of resistant *E. coli* compared to dairy calves, with consistently higher proportions of antimicrobial-resistant isolates, except for AMP. This disparity may be attributed to the transportation process, where young veal calves from different sources are mixed during transit to fattening farms [62,63]. Although, the use of antimicrobial drugs is considered the likely source of AMR, the frequent introductions into groups of young animals from a persistent source as long as the mixing of animal groups that can expose immunologically naïve individuals to AMR have been indicated among the factors implicated in the persistence of AMR on farms [6].

The stressful transportation conditions and intensive rearing environments could heighten veal calves’ susceptibility to infectious diseases, often necessitating antibiotic treatment [35,64]. Notably, common infectious diseases among these animals include neonatal diarrhea and respiratory syndromes, typically treated with antibiotics, like ampicillin, amoxicillin, tetracycline, and sulfamethoxazole/trimethoprim, all antibiotics for which the *E. coli* isolates detected in this study have demonstrated multi-resistance profiles. In these situations, a widespread preventive treatment regimen is often administered to all animals and, as reported in the literature, strains isolated from treated animals exhibit a greater number of resistances and multi-resistances compared to those from untreated counterparts [5,12]. Moreover, the observed increase in resistance among dairy calves during our study period (2017–2022) to antibiotics such as gentamicin (GEN), tetracycline (TET), paromomycin (PRM), and sulfamethoxazole/trimethoprim (SX-T) could be attributed to their high usage in food-producing animals. These antibiotics are often recommended as first or second-line treatments for neonatal diarrhea and calf respiratory diseases, as outlined in the Italian national guidelines for the prudent use of antibiotics in dairy cattle farming [65]. Regarding the critically important antibiotics (CIAs) analyzed in our study, colistin (from the polymyxin class) and enrofloxacin (from the fluoroquinolone class), we did not observe increases in resistance over time, but instead an opposite trend. This observation is further supported by sales data, which indicate a decline in the use of these two antibiotic classes from 5.8 mg/PCU in 2017 to 1.3 mg/PCU in 2022 for quinolones and from 5.2 mg/PCU in 2017 to 0.58 mg/PCU in 2022 for polymyxins [52]. This decline can be attributed to the policies implemented by the Italian National Plan on Antimicrobial Resistance (PNCAR 2017–2021), aimed at reducing the use of CIAs in veterinary medicine.

Regarding MDR, the average number of molecules to which the isolates were multi-resistant was six, and the most prevalent phenotype in both veal and dairy calves was resistant to AMP-ENR-FLO-FLQ-GEN-PRM-TET-SX-T. These values did not deviate from the results reported in the literature, showing four as the mean number of drugs for multidrug resistance, with a median value of five [5]. Comparing the two production types, the percentage of multidrug-resistant isolates was higher in veal calves compared to dairy calves, in all years considered in the study (2017–2022), confirming that *E. coli* isolated in beef farms present higher level of AMR. The isolation of a greater number of multidrug-resistant *E. coli* in veal calves is most probably because these animals receive greater antimicrobial treatments than dairy calves. The comparison of the data on antimicrobial consumption of the Veneto region in 2022 confirms this hypothesis: in veal calves farms the average consumption of antimicrobials, expressed as Defined Daily Doses per Animal (DDDA), was disproportionally higher than that in dairy herds (respectively 35.56 vs. 1.92) [55].

Assessing the temporal trend of multidrug resistance in the Veneto region cattle farms, a significant increase in the percentage of isolated multidrug-resistant *E. coli* in dairy calves was detected: from 66.3% in 2017 to 81.1% in 2022, while in veal calves resistance percentage was high but remained stable throughout the considered years (from 95.2% in 2017 to 96.6% in 2022). This could be related to the fitness advantage of resistant strains in young calves due to their considerable resistance in the external environment and wide adaptation to the calf gut, which allows their persistence for a long time after withdrawal of antimicrobial treatment [11]. Moreover, the administration of milk containing antibiotic residues to the young calves in dairy herds may affect the increase in AMR observed [66]. In a study by Maynou et al. [67], feeding waste milk was associated with beta-lactam and florfenicol resistance but not tetracycline or aminoglycoside resistance. Waste milk has also been identified as a risk factor for ESBL/AmpC-producing *E. coli* positive calves [68]. Although quantitative data was and is still lacking on the impact of feeding waste milk to calves, the European Food Safety Authority carried out a qualitative risk assessment and found that there was an increased risk that the feeding of waste milk to calves would lead to increased fecal shedding of antimicrobial-resistant bacteria [69]. Certainly, further investigations are needed to assess whether this increase of MDR in *E. coli* could be associated with the use of waste milk or other farm management practices.

Eventually, we analyzed the main limitations of this study to allow a critical assessment of the outcomes. The study was performed on a single Italian region over a relatively short period, so specific conditions, such as the type of breeding, cannot mirror other situations. For example, the intensive farming system of veal calves is barely comparable to different kinds of beef breeding, which could affect the AMR level present. Despite the relatively high number of herds included in the study, a potential bias could arise from adopting routine diagnostic ASTs because necroscopy service is required mainly by farms facing mortality problems in calves, which could affect the representativeness. Moreover, the antimicrobials tested were limited to drugs used for cattle therapy without including antimicrobial markers for detection of specific resistance patterns, such as, for example, cefotaxime, which detect the production of Extended Spectrum Beta-Lactamase (ESBL).

## 5. Conclusions

This study highlighted the value of routine ASTs as a tool for the surveillance of AMR in farm animals. The assessment of these data could provide a useful basis for implementing guidelines or establishing recommendations for the responsible and prudent use of antimicrobials at the farm level. Our findings showed a high proportion of antimicrobial-resistant strains for many antimicrobials, including some CIAs, such as quinolones. The study demonstrated a higher level of AMR in veal than dairy calves, probably due to greater antimicrobial treatments in this farming system. Despite the reduction of AMU that occurred in the last five years in Italy, the increase of AMR over time in dairy calves, highlights the need to maintain long-term programs of AMU reduction to task the goal of reducing the AMR level in cattle farms.

## Figures and Tables

**Figure 1 animals-14-01429-f001:**
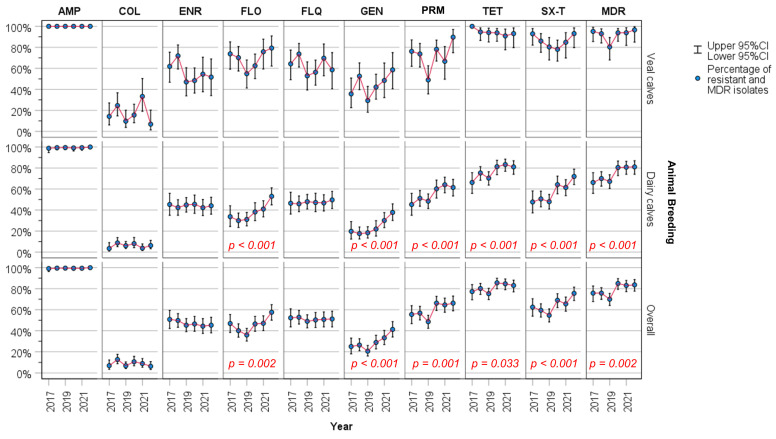
Trend of resistance for each antimicrobial and MDR over the study period (2017–2022) categorized by animal typology (veal and dairy calves) and the overall population. Statistically significant differences are highlight in red. Ampicillin (AMP), colistin (COL), enrofloxacin (ENR), florfenicol (FLO), flumequine (FLQ), gentamicin (GEN), paromomycin (PRM), tetracycline (TET), sulfamethoxazole/trimethoprim (SX-T); multidrug resistant (MDR).

**Figure 2 animals-14-01429-f002:**
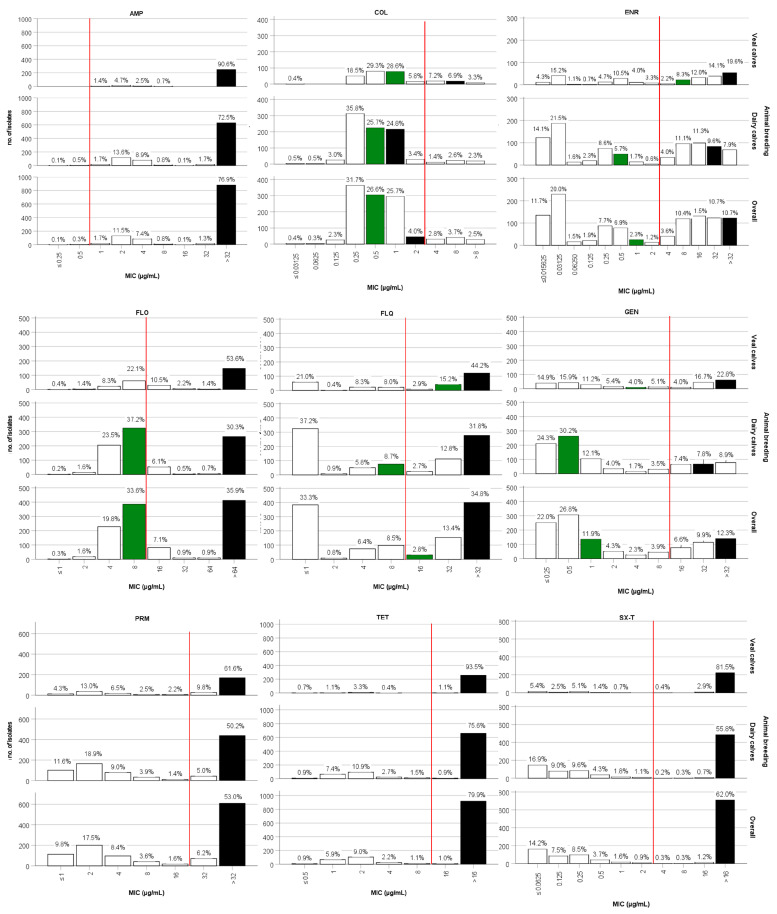
Minimum inhibitory concentration (MIC) distribution (μg/mL) of ampicillin (AMP), colistin (COL), enrofloxacin (ENR), florfenicol (FLO), flumequine (FLQ), gentamicin (GEN), paromomycin (PRM), tetracycline (TET), and sulfamethoxazole/trimethoprim (SX-T). The percentage of isolates is present on top of the bars and the antimicrobial concentrations in μg/mL are displayed on the horizontal axis of the abscissas. Green bars indicate the concentration that inhibits 50% of the isolates (MIC50 values). Black bars indicate the concentration that inhibits 90% of the isolates (MIC90 values). White bars indicate the other MIC values. The red line displays the distribution according to CBPs: isolates to the right of the red line are resistant.

**Table 1 animals-14-01429-t001:** Antimicrobials used with their clinical breakpoint and epidemiological cut-off.

Antimicrobial	Abbreviation	Antimicrobial Class	Clinical Breakpoint (µg/mL)			References and Specie	ECOFF ^d^ (µg/mL)
			S ^a^≤	I ^b^=	R ^c^≥		R ^c^≥
Ampicillin	AMP	Aminopenicillins	0.25	0.5	1	[40], cattle	16
Colistin	COL	Polymyxins	2	-	4	[41], human	4
Enrofloxacin	ENR	Quinolones	0.5	1–2	4	[40], dog	0.25
Florfenicol	FLO	Amphenicols	4	8	16	[40], swine	32
Flumequine	FLQ	Quinolones	4	8	16	[42], all species	4
Gentamicin	GEN	Aminoglycosides	4	8	16	[40], human	4
Paromomycin	PRM	Aminoglycosides	8	16	32	[43], human	n.a.
Tetracycline	TET	Tetracyclines	4	8	16	[40], human	16
Trimethoprim/ sulfamethoxazole	SX-T	Sulfonamides	2/38	-	4/76	[40], human	1

Note: ^a^ susceptible; ^b^ intermediate; ^c^ resistant; n.a.: not available; ^d^ ECOFF provided by EUCAST [44].

**Table 2 animals-14-01429-t002:** Total number of resistant *E. coli* isolates and percentage of resistance for each tested antimicrobial according to Clinical Breakpoints and ECOFFs over the entire study duration. ampicillin (AMP), colistin (COL), enrofloxacin (ENR), florfenicol (FLO), flumequine (FLQ), gentamicin (GEN), paromomycin (PRM), tetracycline (TET), sulfamethoxazole/trimethoprim (SX-T).

Antimicrobials	Animal Breeding	No. of Tested *E. coli* Isolates	Clinical Breakpoint: No. of Resistant *E. coli* Isolates (%)		95%CI	*p*	ECOFF: No. of Resistant *E. coli* Isolates (%)		95%CI	*p*
AMP	veal calves	276	276	(100.0)	-	0.35	250	(90.6)	86.6–93.5	<0.001
	dairy calves	874	869	(99.4)	98.7–99.8		650	(74.4)	71.4–77.1	
	total	1150	1145	(99.6)	99.0–99.8		900	(78.3)	75.8–80.5	
COL	veal calves	276	48	(17.4)	13.4–22.3	<0.001	48	(17.4)	13.4–22.3	<0.001
	dairy calves	874	55	(6.3)	4.9–8.1		55	(6.3)	4.9–8.1	
	total	1150	103	(9.0)	7.4–10.7		103	(9.0)	7.4–10.7	
ENR	veal calves	276	155	(56.2)	50.3–61.9	<0.001	217	(78.6)	73.4–83.0	<0.001
	dairy calves	874	384	(43.9)	40.7–47.2		529	(60.5)	75.2–63.7	
	total	1150	539	(46.9)	44.0–49.8		746	(64.9)	62.1–67.6	
FLO	veal calves	276	187	(67.8)	62.0–73.0	<0.001	158	(57.2)	51.3–62.9	<0.001
	dairy calves	874	328	(37.5)	34.4–40.8		275	(31.5)	28.5–34.6	
	total	1150	515	(44.8)	41.9–47.7		433	(37.7)	34.9–40.5	
FLQ	veal calves	276	172	(62.3)	56.5–67.8	<0.001	217	(78.6)	73.4–83.0	<0.001
	dairy calves	874	414	(47.4)	44.1–50.7		541	(61.9)	58.6–65.1	
	total	1150	586	(51.0)	48.1–53.8		758	(65.9)	63.1–68.6	
GEN	veal calves	276	120	(43.5)	37.8–49.4	<0.001	145	(52.5)	46.6–58.3	<0.001
	dairy calves	874	211	(24.1)	21.4–27.1		257	(29.4)	26.5–32.5	
	total	1150	331	(28.8)	26.2–31.5		402	(35.0)	32.2–37.8	
PRM	veal calves	276	197	(71.4)	65.8–76.4	<0.001	- ^a^	-	-	-
	dairy calves	874	483	(55.3)	51.0–58.5					
	total	1150	680	(59.1)	56.3–61.9					
TET	veal calves	276	261	(94.6)	91.2–96.7	<0.001	261	(94.6)	91.2–96.7	<0.001
	dairy calves	874	669	(76.5)	73.6–79.2		669	(76.5)	73.6–79.2	
	total	1150	930	(80.9)	78.5–83.0		930	(80.9)	78.5–83.0	
SX-T	veal calves	276	234	(84.8)	80.1–88.5	<0.001	236	(85.5)	80.9–89.2	<0.001
	dairy calves	874	499	(57.1)	53.8–60.3		525	(60.1)	56.8–63.3	
	total	1150	733	(63.7)	60.9–66.5		761	(66.2)	63.4–68.8	

^a^ ECOFF not available for PRM.

**Table 3 animals-14-01429-t003:** Resistance patterns of multidrug-resistant *E. coli* isolates based on MIC Clinical Breakpoints. Profiles are reported only if the number of isolates per profile was more than 10. Ampicillin (AMP), colistin (COL), enrofloxacin (ENR), florfenicol (FLO), flumequine (FLQ), gentamicin (GEN), paromomycin (PRM), tetracycline (TET), sulfamethoxazole/trimethoprim (SX-T).

	Veal and Dairy Calves			Veal Calves			Dairy Calves		
Resistance Profile	No. of Isolates within Resistance Profile (%)		Cumulative Percentage	No. of Isolates within Resistance Profile (%)		Cumulative Percentage	No. of Isolates within Resistance Profile (%)		Cumulative Percentage
AMP-ENR-FLO-FLQ-GEN-PRM-TET-SX-T	144	16.0	16.0	66	26.1	26.1	78	12.0	12.0
AMP-ENR-FLO-FLQ-PRM-TET-SX-T	73	8.1	24.1	26	10.3	36.4	47	7.2	19.3
AMP-ENR-FLQ-PRM-TET-SX-T	60	6.7	30.7	9	3.6	39.9	51	7.9	27.1
AMP-PRM-TET-SX-T	57	6.3	37.0	15	5.9	45.8	42	6.5	33.6
AMP-TET-SX-T	48	5.3	42.4	15	5.9	51.8	33	5.1	38.7
AMP-PRM-TET	46	5.1	47.5	2	0.8	52.6	44	6.8	45.5
AMP-ENR-FLQ-PRM-TET	38	4.2	51.7	0	0.0	52.6	38	5.9	51.3
AMP-COL-ENR-FLO-FLQ-GEN-PRM-TET-SX-T	37	4.1	55.8	17	6.7	59.3	20	3.1	54.4
AMP-FLO-PRM-TET-SX-T	31	3.4	59.2	10	4.0	63.2	21	3.2	57.6
AMP-ENR-FLO-FLQ-TET-SX-T	29	3.2	62.4	10	4.0	67.2	19	2.9	60.6
AMP-FLO-GEN-PRM-TET-SX-T	28	3.1	65.5	7	2.8	70.0	21	3.2	63.8
AMP-ENR-FLQ-TET-SX-T	26	2.9	68.4	0	0.0	70.0	26	4.0	67.8
AMP-ENR-FLO-FLQ-GEN-TET-SX-T	22	2.4	70.8	7	2.8	72.7	15	2.3	70.1
AMP-FLO-TET-SX-T	22	2.4	73.3	4	1.6	74.3	18	2.8	72.9
AMP-FLO-TET	15	1.7	74.9	6	2.4	76.7	9	1.4	74.3
AMP-ENR-FLQ-GEN-PRM-TET-SX-T	15	1.7	76.6	3	1.2	77.9	12	1.8	76.1
AMP-FLO-FLQ-GEN-PRM-TET-SX-T	15	1.7	78.3	3	1.2	79.1	12	1.8	78.0
AMP-COL-ENR-FLO-FLQ-PRM-TET-SX-T	13	1.4	79.7	5	2.0	81.0	8	1.2	79.2
AMP-FLO-FLQ-PRM-TET-SX-T	12	1.3	81.0	6	2.4	83.4	6	0.9	80.1
AMP-ENR-FLQ-GEN-PRM-TET	11	1.2	82.3	2	0.8	84.2	9	1.4	81.5
AMP-GEN-PRM-TET-SX-T	11	1.2	83.5	4	1.6	85.8	7	1.1	82.6
Total isolates in reported profiles	753			217			536		
Total isolates in other MDR profiles	149	16.5		36	14.2		113	17.4	
Total MDR isolates	902			253			649		

**Table 4 animals-14-01429-t004:** Antimicrobial susceptibility patterns for the 1150 tested *Escherichia coli* isolates. Ampicillin (AMP), colistin (COL), enrofloxacin (ENR), florfenicol (FLO), flumequine (FLQ), gentamicin (GEN), paromomycin (PRM), tetracycline (TET), sulfamethoxazole/trimethoprim (SX-T).

Antimicrobial	Animal Breeding	Dilution Range (μg/mL)	Minimum MIC Value (μg/mL)	Maximum MIC Value (μg/mL)	MIC_50_ (μg/mL)	MIC_90_ (μg/mL)
AMP	veal calves	0.25–32	1	>32	>32	>32
	dairy calves		≤0.25	>32	>32	>32
	Total		≤0.25	>32	>32	>32
COL	veal calves	0.03125–8	≤0.03125	>8	1	8
	dairy calves		≤0.03125	>8	0.50	1
	Total		≤0.03125	>8	0.50	2
ENR	veal calves	0.015625–32	≤0.015625	>32	8	>32
	dairy calves		≤0.015625	>32	0.50	32
	Total		≤0.015625	>32	1	>32
FLO	veal calves	1–64	≤1	>64	>64	>64
	dairy calves		≤1	>64	8	>64
	Total		≤1	>64	8	>64
FLQ	veal calves	1–32	≤1	>32	32	>32
	dairy calves		≤1	>32	8	>32
	Total		≤1	>32	16	>32
**GEN**	veal calves	0.25–32	≤0.25	>32	4	>32
	dairy calves		≤0.25	>32	0.50	32
	Total		≤0.25	>32	1	>32
PRM	veal calves	1–32	≤1	>32	>32	>32
	dairy calves		≤1	>32	>32	>32
	Total		≤1	>32	>32	>32
TET	veal calves	0.50–16	≤0.50	>16	>16	>16
	dairy calves		≤0.50	>16	>16	>16
	Total		≤0.50	>16	>16	>16
SX-T	veal calves	0.0625–16	≤0.0625	>16	>16	>16
	dairy calves		≤0.0625	>16	>16	>16
	Total		≤0.0625	>16	>16	>16

## Data Availability

The raw data supporting the conclusions of this article will be made available by the authors, without undue reservation.

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
