# Peer review of "Surveillance of Antimicrobial Resistance of Escherichia coli Isolates from Intestinal Contents of Dairy and Veal Calves in the Veneto Region, Northeaster Italy"

_animals, 2024, doi:10.3390/ani14101429_

Round 1
Reviewer 1 Report
Comments and Suggestions for Authors
General comments:
With interest, I have read the emanuscript entitled: “Surveillance on antimicrobial-resistance of Escherichia coli isolates detected from enteric disease in dairy and veal calves in Italy .“ The manuscript offers an important insights but requires refinement in certain areas to enhance clarity, rigour, and overall impact before it is considered for publication. Please find the comments below.
1. Title: Surveillance on antimicrobial-resistance of Escherichia coli isolates detected from enteric disease in dairy and veal calves in Italy
The title of the manuscript can be revised. For example can be improved to: “Surveillance of antimicrobial resistance in Escherichia coli isolated from intestinal contents of veal and dairy calves in Northeastern Italy.”
2. Simple Summary:
· Line 25: what does “routinely available data” refer to? And adding the number 1,150 might not be required here.
3. Abstract:
· Line 35-37: the intended meaning of the sentence is not clear? Does this mean antibiotic resistance levels showed a time-based increase?
4. Introduction:
· The introduction effectively sets the context by discussing the global and veterinary implications of antimicrobial resistance (AMR). However, it could be strengthened by including more specific data or studies related to AMR trends in Italy, particularly in veal and dairy calves.
· The manuscript briefly mentions the study aims but does not clearly articulate specific hypotheses or research questions.
· The introduction lacks consistency and coherence, failing to adequately outline the research hypothesis or clarify the specific research question the study intends to address.
5. Materials and Methods:
· What breed of dairy cattle were included in the study?
- What criteria were used to select the antimicrobials included in this study for the antimicrobial resistance testing? The manuscript could benefit from a more detailed explanation regarding the choice of antimicrobials included for resistance testing. What criteria were used to select these antimicrobials? Are they commonly used in veal and dairy calf production in the region, or are there specific resistance concerns that motivated their selection?
6. Results
· Line 205: animals or cattle?
· Avoid using words such as always( line 248) and entire (line 246), as they are too generic.
· Using a plot, it would be great if you could add the temporal distribution of minimum inhibitory concentration (MIC) for each antimicrobial tested.
7. Discussion
· The first paragraph and parts of the second paragraph of the discussion section could be taken into the introduction part to justify the importance of selecting E. coli and AMR categorization methods used in the study.
Line 352: intended meaning is not clear.
- The manuscript would benefit from an indication of its limitations. For example, potential biases introduced by the sample or geographical area selection or limitations in the representativeness of the study sample could be discussed.
The overall quality of English in the manuscript is good, with minor grammatical errors.
Author Response
General comments:
With interest, I have read the manuscript entitled: “Surveillance on antimicrobial-resistance of Escherichia coli isolates detected from enteric disease in dairy and veal calves in Italy”. The manuscript offers an important insights but requires refinement in certain areas to enhance clarity, rigour, and overall impact before it is considered for publication. Please find the comments below.
Answer: Dear Reviewer, on behalf of all the authors, I would like to express our sincere gratitude to you for appreciating our manuscript.
1.Title: Surveillance on antimicrobial-resistance of Escherichia coli isolates detected from enteric disease in dairy and veal calves in Italy
The title of the manuscript can be revised. For example can be improved to: “Surveillance of antimicrobial resistance in Escherichia coli isolated from intestinal contents of veal and dairy calves in Northeaster Italy.”
Answer: The Authors agree with the Reviewer’s comments, and the title of the manuscript has been modified to “Surveillance on antimicrobial-resistance of Escherichia coli isolates from enteric intestinal contents of dairy and veal calves in the Veneto region, Northeaster Italy”, also considering the suggestion of another reviewer.
2.Simple Summary:
Line 25: what does “routinely available data” refer to? And adding the number 1,150 might not be required here.
Answer: The Authors agree with the Reviewer’s comments, and the sentence at line 25 has been changed. The number of E. coli strains tested has been deleted.
3.Abstract:
Line 35-37: the intended meaning of the sentence is not clear? Does this mean antibiotic resistance levels showed a time-based increase?
Answer: The Reviewer's interpretation is correct. The sentence has been modified to make it clear.
4.Introduction:
The introduction effectively sets the context by discussing the global and veterinary implications of antimicrobial resistance (AMR). However, it could be strengthened by including more specific data or studies related to AMR trends in Italy, particularly in veal and dairy calves.
The manuscript briefly mentions the study aims but does not clearly articulate specific hypotheses or research questions.
The introduction lacks consistency and coherence, failing to adequately outline the research hypothesis or clarify the specific research question the study intends to address.
Answer: The introduction has been revised according to address the Reviewer's comment. In particular, more specific data regarding AMR trends in Italy have been included (L. 70-79). Two references have also been added (Formenti et al., 2021; Ferroni et al., 2022).
The objectives of the study have also been revised, outlining the research hypotheses (L. 134-138).
- Materials and Methods:
What breed of dairy cattle were included in the study?
Answer: The breed of dairy cattle was included in the chapter Materials and Methods (L. 162-163).
What criteria were used to select the antimicrobials included in this study for the antimicrobial resistance testing? The manuscript could benefit from a more detailed explanation regarding the choice of antimicrobials included for resistance testing. What criteria were used to select these antimicrobials? Are they commonly used in veal and dairy calf production in the region, or are there specific resistance concerns that motivated their selection?
Answer: A more detailed description of the criteria used for the selection of antimicrobial has been added in the chapter Materials and Methods, subchapter 2.3. Antimicrobial susceptibility testing (L. 217-223).
- Results
Line 205: animals or cattle?
Answer: The word “animals” was replaced with “calves” so as to make the meaning of the sentence clearer (L. 256).
Avoid using words such as always (line248) and entire (line 246), as they are too generic.
Answer: The authors agree with the Reviewer’s comment and in order to address this comment sentences were rephrased more specifically (L. 43; 269; 275; 292).
Using a plot, it would be great if you could add the temporal distribution of minimum inhibitory concentration (MIC) for each antimicrobial tested.
Answer: As suggested by the Reviewer, figures have been added (supplementary material, Figures S1) where the temporal distributions of the mean and median MICs value for each antimicrobial over the period 2017-2022) are reported. The figure is cited in the text at L 365. Any correlation between year and MICs value was also assessed. Accordingly, the paragraph Statistical analysis (L. 248-249 has been modified. The results of this analysis were reported at L. 345, L. 3456.
- Discussion
The first paragraph and parts of the second paragraph of the discussion section could be taken into the introduction part to justify the importance of selecting E. coli and AMR categorization methods used in the study.
Answer: As recommended by the Reviewer, the first paragraph of the discussion has been moved into the introduction (L. 75-84).
Line 352: intended meaning is not clear.
Answer: The sentence has been erased and replaced by a new sentence to clarify the meaning of the sentence (409-413).
The manuscript would benefit from an indication of its limitations. For example, potential biases introduced by the sample or geographical area selection or limitations in the representativeness of the study sample could be discussed.
Answer: We thank the reviewer for his suggestion. The limits of the study were briefly discussed (L. 555-566).
Reviewer 2 Report
Comments and Suggestions for Authors
The paper’s authors have done a surveillance in the Veneto region in Italy. They have calculated the proportion of antimicrobial resistant (AMR) E. coli isolated from enteric disease claves for nine antibiotics, investigating the potential differences in the SAMR between dairy and veal calves and identifying potential AMR variations over the time. Their findings demonstrated a high proportion in AMR, higher in veal then in dairy claves and in general the AMR has increased through the time. In general, the paper is a well designed, executed and written paper. However, there are some details that can improved the already good paper.
In general, I think that the use of prevalence has to be rethink, in most of the paper should be better use the term proportion. I have taken the liberty to add some extra information related to that. Both terms, "proportion of antimicrobial resistances" and "prevalence of antimicrobial resistances," are relevant and useful in the context of antimicrobial resistance, but they may have slightly different nuances in their interpretation.
Proportion of antimicrobial resistances: This term refers to the proportion or fraction of tested bacterial strains that exhibit resistance to a specific antibiotic. For example, if out of 100 bacterial strains analyzed, 30 are resistant to an antibiotic, then the proportion of antimicrobial resistances for that particular antibiotic would be 30%. This metric provides a direct insight into the magnitude of resistance in relation to the total number of strains analyzed.
Prevalence of antimicrobial resistances: Prevalence refers to the proportion of a population that exhibits antimicrobial resistance at a given time. This measure can be based on data collected from patient samples, clinical settings, or environmental settings. For example, if 500 patient samples are examined in a hospital and it is found that 100 exhibit antimicrobial resistance, then the prevalence of antimicrobial resistances in that hospital would be 20%. Prevalence offers a broader perspective on the extent of resistance in a particular population.
The proportion of antimicrobial resistances focuses on the relationship between resistant strains and the total number of strains analyzed, while the prevalence of antimicrobial resistances focuses on the proportion of the population affected by resistance at a given time.
I think that you have the data to calculate the prevalence and seems that it was an original goal, as it is mention in the text (L29 and L105), but in the results you do not show this result.
Specific comments:
L3: Title: “Surveillance on antimicrobial-resistance of Escherichia coli isolates detected from enteric disease in dairy and veal calves in Italy”, add the Veneto region to the title.
L35-37: In the abstract there is no need to use the acronyms for the antibiotics.
L120: were collected from, instead of were collected by.
L150: Table 1: Move the ECOFF of SX-T a little bit down.
L159-161: How many microL of the bacterial suspension did you add to how many ml of the CAM-HB medium?
L204: specify that the calves had enteritis.
L208: The calculation of the prevalence can be added after this paragraph.
L211-215: Add the SX-T information. This sentence has to be rephrased as the exceptions are bigger (5 antibiotics) that what you find normal (4 antibiotics have CBPs values greater that the ECOFFs.
L240: Add a space between During and the.
L255: Add MIC to the title.
L349: Salmonella enterica serotype Choleraesuis, instead of Salmonella Choleraesuis.
L493: Check if the use of prevalence here is correct based on what I mention before.
Author Response
The paper’s authors have done a surveillance in the Veneto region in Italy. They have calculated the proportion of antimicrobial resistant (AMR) E. coli isolated from enteric disease claves for nine antibiotics, investigating the potential differences in the SAMR between dairy and veal calves and identifying potential AMR variations over the time. Their findings demonstrated a high proportion in AMR, higher in veal then in dairy claves and in general the AMR has increased through the time. In general, the paper is a well designed, executed and written paper. However, there are some details that can improved the already good paper.
Answer: The Authors thank the Reviewer for the positive comments on the paper and for the suggestions provided.
In general, I think that the use of prevalence has to be rethink, in most of the paper should be better use the term proportion. I have taken the liberty to add some extra information related to that. Both terms, "proportion of antimicrobial resistances" and "prevalence of antimicrobial resistances," are relevant and useful in the context of antimicrobial resistance, but they may have slightly different nuances in their interpretation.
Proportion of antimicrobial resistances: This term refers to the proportion or fraction of tested bacterial strains that exhibit resistance to a specific antibiotic. For example, if out of 100 bacterial strains analyzed, 30 are resistant to an antibiotic, then the proportion of antimicrobial resistances for that particular antibiotic would be 30%. This metric provides a direct insight into the magnitude of resistance in relation to the total number of strains analyzed.
Prevalence of antimicrobial resistances: Prevalence refers to the proportion of a population that exhibits antimicrobial resistance at a given time. This measure can be based on data collected from patient samples, clinical settings, or environmental settings. For example, if 500 patient samples are examined in a hospital and it is found that 100 exhibit antimicrobial resistance, then the prevalence of antimicrobial resistances in that hospital would be 20%. Prevalence offers a broader perspective on the extent of resistance in a particular population.
The proportion of antimicrobial resistances focuses on the relationship between resistant strains and the total number of strains analyzed, while the prevalence of antimicrobial resistances focuses on the proportion of the population affected by resistance at a given time.
I think that you have the data to calculate the prevalence and seems that it was an original goal, as it is mention in the text (L29 and L105), but in the results you do not show this result.
Answer: We thank the reviewer for his comments and agree with the definitions of “Proportion of antimicrobial resistances” and “Prevalence of antimicrobial resistances”.
In this study, we examined a single strain of E. coli isolated from each calf over a time period of 6 years. In addition, we examined fecal samples from animals with enteric pathology submitted for diagnostic purposes to our laboratory. It is therefore likely that these animals cannot be considered a representative sample of beef and dairy calves reared in the Veneto region because no randomized sampling was performed.
Accordingly, in agreement with the definitions provided by reviewer, we think it is more correct to use the definition “Proportion of antimicrobial resistances” instead of “Prevalence of antimicrobial resistances”. In fact, we evaluated the proportion of tested E. coli strains that exhibit resistance to a specific antibiotic.
Therefore , we have uniformed the definition throughout the paper.
Specific comments:
L3: Title: “Surveillance on antimicrobial-resistance of Escherichia coli isolates detected from enteric disease in dairy and veal calves in Italy”, add the Veneto region to the title.
The Authors agree with the Reviewer’s comments, and the title of manuscript been changed in “Surveillance on antimicrobial-resistance of Escherichia coli isolates from enteric intestinal contents of dairy and veal calves in the Veneto region, Northeaster Italy”, also considering the suggestion of another reviewer.
L35-37: In the abstract there is no need to use the acronyms for the antibiotics.
Answer: We agree with the Reviewer's comment. In the Abstract, acronyms have been deleted.
L120: were collected from, instead of were collected by.
Answer: We agree with the Reviewer's comment. The sentence has been modified accordingly (L160).
L150: Table 1: Move the ECOFF of SX-T a little bit down.
Answer: The table has been modified as suggested.
L159-161: How many microL of the bacterial suspension did you add to how many ml of the CAM-HB medium?
Answer: The requested information were added to the sentence (L. 200-201).
L204: specify that the calves had enteritis.
We agree with the Reviewer's comment. The sentence has been modified accordingly (L. 254).
L208: The calculation of the prevalence can be added after this paragraph.
Answer: Please see previous response regarding Proportion/Prevalence of antimicrobial resistances.
L211-215: Add the SX-T information. This sentence has to be rephrased as the exceptions are bigger (5 antibiotics) that what you find normal (4 antibiotics have CBPs values greater that the ECOFFs.
Answer: We apologize for the mistake. The sentence has been rephrased (L. 263-266).
L240: Add a space between During and the.
Answer: The sentence has been modified as requested by another reviewer (L. 292).
L255: Add MIC to the title.
Answer: We apologize for the mistake. The title has been corrected.
L349: Salmonella enterica serotype Choleraesuis, instead of Salmonella Choleraesuis.
Answer: We apologize for the mistake. "enterica serotype" was added.
L493: Check if the use of prevalence here is correct based on what I mention before.
Answer: We replaced “prevalence” with “proportion of antimicrobial resistant strains”. Please see previous response regarding Proportion/Prevalence of antimicrobial resistances.